# A New Rotary Magnetorheological Damper for a Semi-Active Suspension System of Low-Floor Vehicles

Yu-Jin Park [1], Byung-Hyuk Kang [2] and Seung-Bok Choi [2,3,*]

1 Korea Initiative for Fostering University of Research and Innovation, Inha University, Incheon 21999, Republic of Korea; eugene5059@inha.ac.kr
2 Department of Mechanical Engineering, The State University of New York, Korea (SUNY Korea), Incheon 21985, Republic of Korea; 1357op@gmail.com
3 Department of Mechanical Engineering, Industrial University of Ho Chi Minh City (IUH), Ho Chi Minh City 700000, Vietnam
* Correspondence: seungbok.choi@sunykorea.ac.kr

**Abstract:** This study explores the significance of active suspension systems for vehicles with lower chassis compared to conventional ones, aiming at the development of future automobiles. Conventional linear MR (magnetorheological) dampers were found inadequate in ensuring sufficient vibration control because the vehicle's chassis becomes lowered in the unmanned vehicles or purposed-based vehicles. As an alternative, a rotary type of MR damper is proposed in this work. The proposed damper is designed based on prespecified design parameters through mathematical modeling and magnetic field analyses. Subsequently, a prototype of the rotary MR damper identical to the design is fabricated, and effectiveness is shown through experimental investigations. In configuring the experiments, a proportional-integral (PI) controller is employed for current control to reduce the response time of the damper. The results presented in this work provide useful guidelines to develop a new type of MR damper applicable to various types of future vehicles' suspension systems with low distance from the tire to the body floor.

**Keywords:** magnetorheological fluid; rotary MR damper; low-floor vehicles; finite element analysis; magnetic analysis; response time; damping force





## 1. Introduction

It is well known that vehicle suspension systems can be classified into the passive, the semi-active, and active types. Recently, many studies have been conducted about the semi-active suspension system due to its enhanced ride comfort and road holding with fail-safe capability compared to the other two types. Especially, since magnetorheological (MR) fluid has been applied to several types of dynamic systems to control unwanted vibrations, more attention and research has been conducted on the semi-active suspension of the vehicles. These numerous works could bring both MR fluid itself and MR damper for the vehicle suspension system to the corresponding market as commercial outcomes [1,2]. In fact, there are many types of MR dampers applicable to vehicle suspension systems: mono-tube type with and without the bypass hole, twin-tube type with the single-end, mono-tube type with double-end, mono-tube type with the external magnetic circuit core, and pinch node type [3–6]. Each type has advantages and disadvantages over the other types. For example, the mono-tube with the single-ended MR damper produces the maximum damping force at same conditions, while the mono-tube type with the external core MR damper has the largest stroke motion during driving operation. However, most of the MR dampers proposed or developed for vehicle suspension systems so far are the linear types to provide the translation stroke in the direction of up and down. This large stroke motion in a vertical direction is possible since the space (or height) from the wheel to the floor of the car body is sufficient (around ±30 mm) for the installation of the linear MR damper vertically. Figure 1

shows a schematic configuration of the typical linear MR damper, which consists of the MR piston assembly, gas chambers, and magnetic circuits cores.

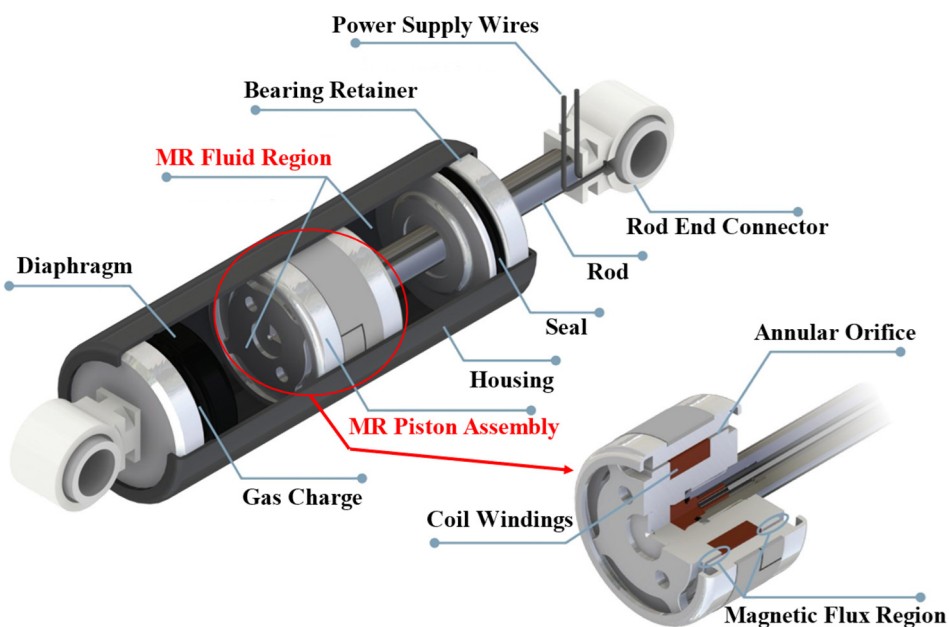

**Figure 1.** Schematic configuration of the linear MR damper.

As for the research on rotary MR dampers, Giorgetti et al. [7] firstly introduced a rotary MR damper applicable to the front-wheel suspension of a compact car. Some advantages of the proposed rotary MR damper, such as the small quantity of fluid and the low abrasion of the seals, were discussed, followed by a swinging type of the rotary MR damper that was designed to reduce the space requirement and mechanical resistance of the heavy off-road profiles. After analyzing the flow motion of the annular duct, several principal design parameters were determined. The operational angle is chosen by 60 deg to reproduce all the movements of the previous suspension motion. It has been shown from measurements that the damping torque of the rotary MR damper can cover the damping range of the conventional linear MR damper. It is noted here that verification of the results presented in this work is not possible since the information about many design parameters and magnetic analysis are not presented. Lee et al. [8] designed a rotary MR damper for an unmanned vehicle suspension system and evaluated the field-dependent torque performance through the finite element method and an experimental test. The damping torque at a disk rotating speed of 10 m/s was obtained by 239.2 and 576.78 Nm at the current 0.5 A and 1.5 A, respectively. However, in this work, the limit of the rotational angle and the height of the MR damper did not consider the design. Thus, the operating principle of the MR damper studied in this work is the same as MR clutch in which the torque is the resistance of the servo motor to overcome a reduction in motor speed due to the increment of the apparent viscosity of the MR damper with respect to the current applied to it. Imaduddin et al. [9] published a review article of a rotary MR damper, emphasizing the design and modeling methods. It has been mentioned that despite successful implementation of linear MR dampers to the semi-active suspension systems of variable vehicles, there are still limitations to be resolved, for example, the linear MR damper needs a large space to install (especially height), there is an inability to accommodate high wheel travel due to the risk of the buckle in the damper, and there is a large amount of MR fluid and less resistance to damage from foreign objects. Two types of the rotary MR damper with continuous angle and limited angle were also discussed. The configuration of the continuous type is the same as the disk-type of MR clutch and MR brake, which has a relatively large design to achieve high damping torque. The limited angle type is operated by the vane with an inner magnetic coil valve, and the rotational angle is determined by the movement of

the rotating hub connected to the vane. The vane type of rotary MR damper has several benefits, including low height, a compact side without any chamber, a small amount of MR fluid, and a combination of flow and shear mode generation by the rotational angle. However, a lot of challenging effort is still required for successful realization in practice. For example, the optimal design of the non-cylindrical shape of the vane is one of the most serious problems to be resolved. Imaduddin et al. [10] also proposed a bypass rotary MR damper for the semi-active suspension system of conventional vehicles. In this design, the rotational angle was limited by 70 degrees, and the vane size was fixed at $40 \times 40$ mm with a radius of the vane center of 4.5 mm. The outer diameter of the rotary MR damper was determined to be 50 mm to locate three magnetic cores at the center line. From the simulation, the proposed rotary MR damper produced more than 1000 Nm with 1 A and 1.5 rad/s. It should be noted that these results were obtained without any magnetic analysis and operating angle condition.

Sapinski [11] investigated an energy harvesting system utilizing a rotary motion obtained from three components: a rotary MR damper to vary the damping characteristics, a rotary power generator producing electrical power, and a conditioning electronics unit to interface with the damper and generator. This work is similar to the self-powered linear MR damper using the energy harvesting system, in which the linear MR damper is replaced by the rotary MR damper with the rotary generator. In this work, a multi-disk type of the rotary MR damper is made and connected with the generation. It has been shown that the energy recovered by the generator is sufficient to operate the rotary MR damper, but the conditioning electronics unit needs to be changed so that the harvested voltage is less than the maximum limit. Yu et al. [12] proposed a rotary MR damper consisting of two driven disks and an active rotary disk to achieve an output with a large stiffness change. In the performance evaluation via experimental test, the effects of the current, angle, and frequency on the torque–angle loops and torque–frequency loops are considered. The height and width of the MR damper were chosen as 57.6 mm and 87 mm, respectively. It has been shown that the torque is increased as the angle increases, and the dynamic range is increased as the current increases, as expected. It was also observed from experimental results that the maximum torque was identified by 10 Nm. Such a low level of torque is much less than the torque of a traditional linear MR damper, and hence the proposed rotary MR damper does not provide any practical feasibility. It is noted here that expect the rotary MR damper, the rotational motion devices utilizing MR fluid have been worked a lot, focusing on the brake, clutch, bearing, and shaft vibration control [13–17].

Recently, diverse configurations of future vehicles are introduced in several places, such as CES 2024, http://www.ces.tech; accessed on 27 March 2024 [18], held in January this year at Las Vegas. At this technology show, several cars and concept cars were introduced by car makers and electronics companies. It is very interesting to see that most of the future cars for the unmanned mobility city (UMC) or the purposed-based vehicles (PBV) for a special mission have a very low distance (height) from the wheel to the floor of the vehicle body. Therefore, a commercially available linear MR damper cannot be installed in the limited space. Figure 2 shows the height between the wheel and the body floor of the car for the conventional and future vehicles, respectively. The photo shown in Figure 2b of the future vehicle is one of the PBVs presented at CES 2024. It is clearly seen that there is a limited space to install an MR damper in a vertical direction. Therefore, a new type of rotary MR damper, which is applicable to the low-floor vehicle suspension systems, is required.

The amount of research on the rotary MR damper for vehicle suspension systems is much less than for the conventional linear MR damper. It is identified from the above literature survey on the rotary MR damper that there is no study to meet the prescribed targets of design parameters and damping forces. Most of the rotary MR dampers have been designed without the specified target performances. Consequently, the main technical contribution of this work is to design and manufacture a prototype of a rotary MR damper and demonstrate its effectiveness in terms of the possibility of practical application to the future vehicles featuring a very low-floor height. In this work, a vane type of rotary MR

damper is designed to resolve the strict space limitation. The flow motion of MR fluid is then limited by the rotational angle range of the vane, where both flow mode and shear mode occur. After analyzing the governing equations and magnetic flux distribution, the principal design parameters to achieve the prescribed targets are determined. The size and number of magnetic circuit cores are determined through the magnetic flux density of the MR damper associated with the design parameters. Then, a protype is manufactured and tested to achieve the field-dependent damping force. It is noted here that the design method and results presented in this work will be useful guidelines to make an efficient rotary MR damper for specialized vehicles to appear in UMC vehicles.

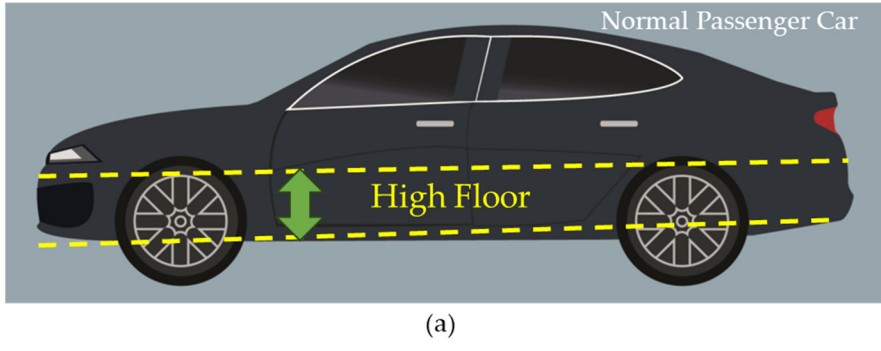

(a)

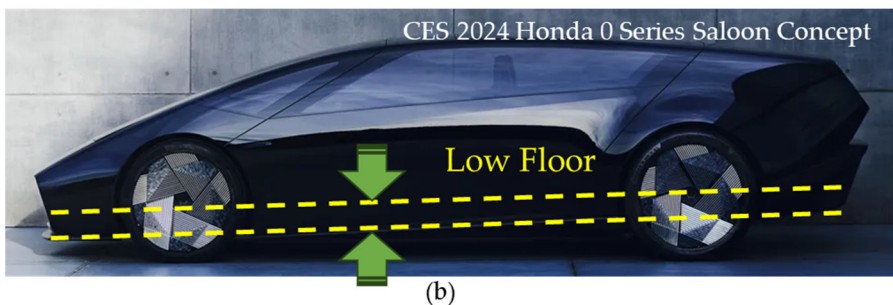

(b)

**Figure 2.** Height from the tire to the body floor: (**a**) conventional vehicle, (**b**) future car.

## 2. Configuration and Modeling

As mentioned in the Introduction, in this work, desired targets (or specifications) are firstly set up to design a rotary MR damper. Since the proposed MR damper is to be installed a small-sized future car, the desired specifications are chosen as follows: (i) maximum damping torque target: 600 Nm @ 30 m/s or higher, (ii) response time for current control: within 50 ms @ 90% rise time criterion, (iii) operating angle range by the vane: $\pm 30$ degree, (iv) maximum diameter: 150 mm, (v) maximum width: 190 mm. In order to meet these targets, several configurations of the rotary MR damper are first investigated and created. Figure 3 presents two different types of the rotary MR damper. The first configuration depicted in Figure 3a is designed to generate damping torque by inducing rotational motion on the upper arm, thereby facilitating the flow of the MR fluid. This configuration offers the advantage of relatively straightforward construction and a wide range of rotational motion capabilities. However, it necessitates careful attention to detail in the design of the oil packing to prevent potential oil leakage between the MR damper and the cylinder contact. Moreover, the inclusion of a bearing guide with a sloped surface is essential for its operation. Moreover, the manufacturing process for this configuration is complex. On the other hand, the second damper structure shown in Figure 3b offers distinct characteristics and potential advantages. The flow motion of MR fluid through the orifice is easily brought about by the rotor and convenient to make as a compact size. In addition, the intensity of the magnetic field can easily control the resistance of fluid flow and hence the field-dependent damping force. Therefore, in this work, the configuration shown in Figure 3b is chosen and modified to achieve the specified targets.

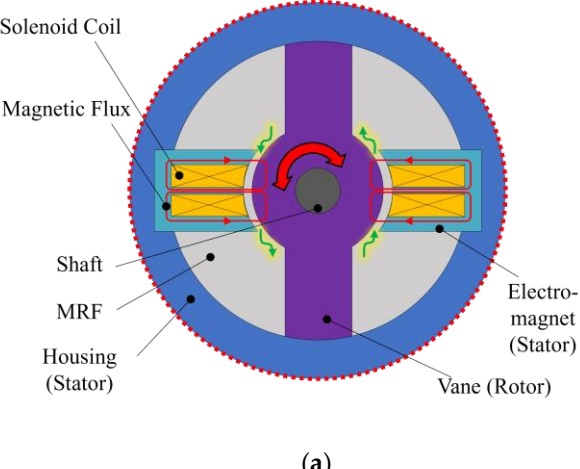

(**a**)

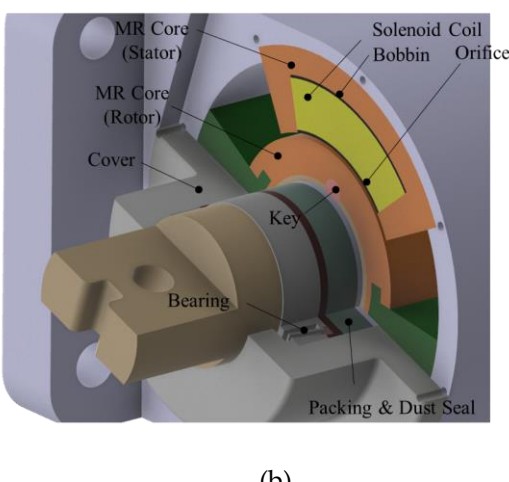

(**b**)

**Figure 3.** Configurations of the rotary MR damper: (**a**) cylinder type, (**b**) key-home type.

In this work, the configuration shown in Figure 3b is modified a little by adding the bypass holes to reduce the damping coefficient in the low frequency (low rotational speed) to acquire a good ride comfort from the start. The modified configuration is depicted in Figure 4. This figure is useful to drive the pressure drops proportional to rotational speed and pressure drop proportional to input current. The pressure drop due to the viscosity (dynamic viscosity) of MR fluid, and the pressure drop due to the yield stress of MR fluid is given by Equations (1)–(4) [19,20]:

$$P_{vis} = \frac{12\eta L}{wh^2}\left(\frac{1}{h} - \frac{2D_r}{D_v^2 - D_r^2}\right)\cdot Q \tag{1}$$

$$Q = \frac{D_v^2 - D_r^2}{8}w\cdot\omega \tag{2}$$

$$P_y = c\frac{L_p}{h}\big|\tau_y(H)\big|\cdot sgn(\omega) \tag{3}$$

$$c = 2.07 + \frac{12\eta|Q|}{12\eta|Q| + 0.4wh^2\big|\tau_y(H)\big|} \tag{4}$$

$$\Delta P = P_{vis} + P_y \tag{5}$$

$$T = \frac{D_v^2 - D_r^2}{4}w\cdot\Delta P \tag{6}$$

In the above equations, $P_{vis}$ is the pressure loss for rotor speed, $P_y$ is the pressure loss for magnetic field, $Q$ is the total flow, $\Delta P$ is the total pressure drop, $\eta$ stands for the viscosity coefficient of the MR fluid, $L$ represents the length inside the core, $w$ indicates the total height of the damper, $D_r$ denotes the outer diameter of the MR core, and $D_v$ represents the outer diameter of the vane rotor. $h$ denotes the gap size, $c$ is the flow velocity profile function, $\tau_y(H)$ is field-induced or field-dependent yield stress of the MR fluid, $H$ denotes the magnetic field intensity, $w$ symbolizes the angular velocity, $T$ refers to the static torque, and $sgn(\omega)$ represents the signum function with respect the angular speed. It is identified that the pressure drop proportional to rotational speed is derived in Equations (1)–(3), and the flow rate $Q$ is expressed similarly to Equation (1) as it occurs between the rotor and housing. The field-dependent pressure drop is dependent on the yield stress of MR fluid, which is a function of the current, as shown in Equation (4), and $c \approx 2.07 + 1/(1 + 0.4\mathcal{T})$ bounded to the interval [2.07, 3.07] [21]. The total pressure drop can be expressed as the sum of pressure drop proportional to rotational speed and pressure drop proportional to input current, as shown in Equation (5). Pseudo-static rotational force represents the static force or moment exerted on the rotating object. By utilizing the total pressure drop obtained earlier, the pseudo-static rotational force can be expressed in terms of torque as given in Equation (6). Since the proposed rotary MR damper has a bypass hole, the flow direction inside the damper is indicated by the green line in Figure 4. Under the assumption of neglecting frictional forces, the total flow can be considered as the sum of the flow through the orifice and the flow through the bypass:

$$Q = Q_0 + Q_b \tag{7}$$

However, it can be assumed that the pressure drops inside are all proportional; thus, the pressure drop can be represented by Equation (7). Now, Equations (8)–(10) are derived, respectively, for the pressure drop relationships of the orifice and the bypass hole. Equation (11) is obtained using the Hagen–Poiseuille equation, which is a common formula used to calculate pressure drop caused by fluid flow inside a pipe. However, difficulties could arise in calculating the pressure drops due to the ambiguity in determining the flow rates. In this work, in order to resolve this, the pressure drop analysis is carried out using dimensionless analysis (non-dimensionalization).

$$\Delta P = \Delta P_0 = \Delta P_b \tag{8}$$

$$\Delta P_0 = \frac{12\eta L}{wh^2}\left(\frac{1}{h} - \frac{2D_r}{D_v^2 - D_r^2}\right) \cdot Q_0 + c\frac{L_p}{h}\left|\tau_y(H)\right| \cdot sgn(\omega) \tag{9}$$

$$where\ c = 2.07 + \frac{12\eta|Q_0|}{12\eta|Q_0| + 0.4wh^2\left|\tau_y(H)\right|} \tag{10}$$

$$\Delta P_{bypass} = \frac{128\eta L_b}{\pi D_b^4 N_b} \cdot (Q - Q_0) \tag{11}$$

Figure 5 depicts a schematic of fluid flow within the parallel plate utilized for deriving dimensionless flow rates [21]. In the above, $\tau_y$ is the yield stress, $h_1$ is the region II's height, $h_2$ is the region I's height, $D_0$ is the gap total (orifice), $L_p$ is the magnetic pole length of the orifice hole, $D_b$ is the gap of the bypass, $N_b$ is the number of bypass holes, $L_b$ is the magnetic pole length of the bypass hole. It shows the flow for both the orifice and bypass individually, with the nomenclature for each location within the plate provided on the right. The dimensionless flow rate for the damper orifice is initially derived. Here, the key dimensionless variables, $Q^*$, $T^*$, $L^*$, and $V^*$ represent the flow rate, the static the torque, the length, and the fluid velocity, respectively. They are to be utilized significantly, and the expression can sufficiently represent Equation (7). Having derived Equation (12), the relationship for the bypass can be deduced from the equations governing the bypass flow rate and pressure drop. Thus, the following equations are obtained:

$$Q^* = 1 - \frac{3L^*V^*}{1-(0.4-L^*)T^*} - \frac{(1.4+L^*)T^*}{1+(0.4+L^*)T^*}$$

$$\begin{cases} Q^* = \frac{12\eta L Q_0}{wh^3 \Delta P}, \ V^* = \frac{2\eta Lv}{h^2 \Delta P}, \ T^* = \frac{2L_p|\tau_y|}{h\Delta P}, \ L^* = \frac{L_p}{L} \\ 0 < Q^* < 1, \ 0 < T^* < 1, \ 0 < L^* < 1 \\ 0 < V^* < \frac{(1-(1-L^*)Q^*-T^*)^2}{L^*(1-(1-L^*)Q^*)} \end{cases} \tag{12}$$

Now, using Equation (7), the proportional relationship between the orifice and bypass leads to the derivation of equations governing the flow rate and pressure drop. The derived equation is presented by Equations (13)–(15). In this equation, $a_1$ represents the flow rate expression for the bypass, and $a_2$ denotes the relationship associated with the bypass:

$$a_1 \frac{Q}{\Delta P} = \frac{a_1}{a_2} + Q^* = 1 + \frac{a_1}{a_2} - \frac{3L^*V^*}{1-(0.4-L^*)T^*} - \frac{(1.4+L^*)T^*}{1+(0.4+L^*)T^*} \tag{13}$$

$$a_1 = \frac{12\eta L}{wh^3} \tag{14}$$

$$a_2 = \frac{128\eta L_b}{pi D_b^4 N_b} \tag{15}$$

Therefore, summarizing the dimensionless pressure drop relationship is given by the following equation:

$$P^* = \frac{a_2}{a_1+a_2}\left(1 + \frac{3L^*V^{*+}P^*/6}{P^*-(0.4-L^*)T^*} + \frac{(1.4+L^*)L^*T^{*+}P^*}{P+(0.4+L^*)L^*T^*}\right) \tag{16}$$

$$P^* = \frac{wh^3 \Delta P}{12\eta LQ}, \ V^{*+} = \frac{whv}{Q}, \ T^{*+} = \frac{wh^2|\tau_y|}{6\eta Q}, \ L^* = \frac{L_p}{L} \tag{17}$$

$$\Delta P = \begin{cases} a_1 P^* Q & 2L_p|\tau_y| < a_2 h|Q| \\ a_2 Q & 2L_p|\tau_y| \geq a_2 h|Q| \end{cases} \tag{18}$$

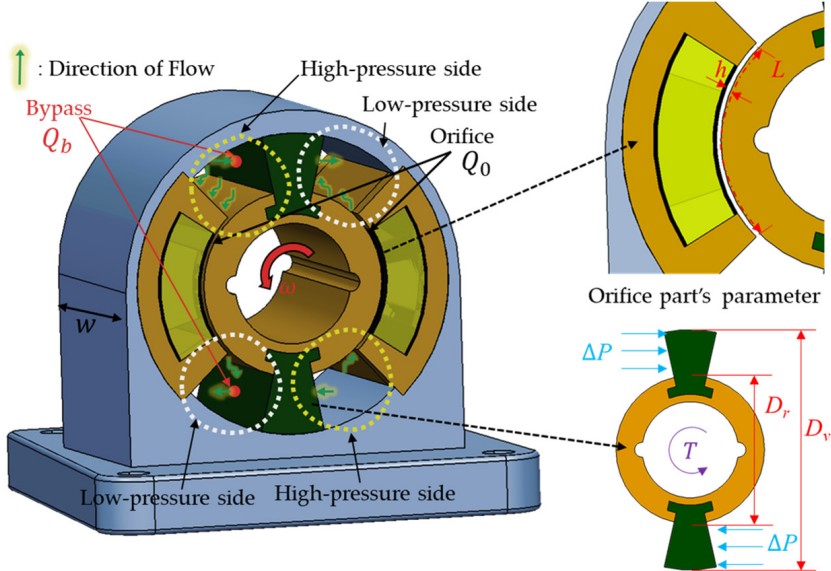

**Figure 4.** The schematic for flow direction and pressure drop of the rotary MR damper.

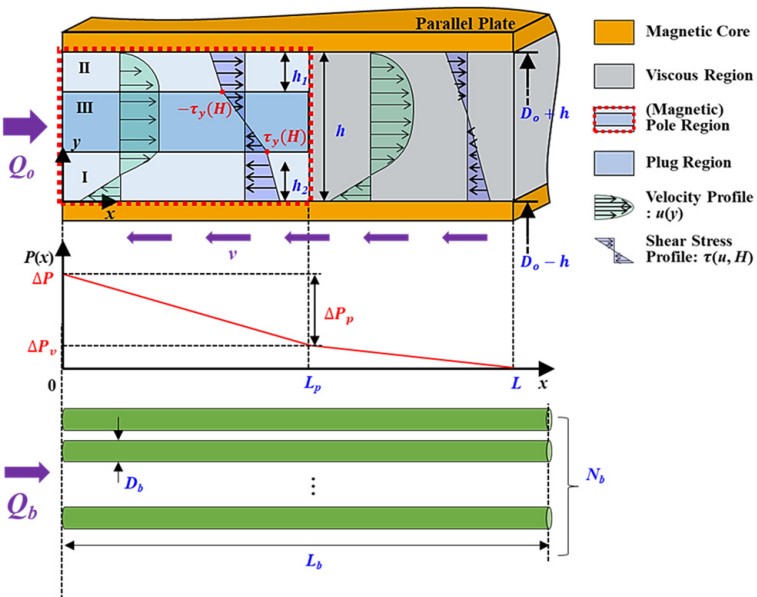

**Figure 5.** Non-dimensional flow rate of the rotary MR damper using the parallel plate.

## 3. Magnetic Analysis and Design Parameters

Before determining the principal design parameters to meet the prespecified targets, the magnetic field distribution of the rotary MR damper needs to be undertaken. In this work, through the finite element analysis (FEA) simulations, the design elements are analyzed in terms of the rotational damping relative to the input current. Additionally, computational fluid dynamics (CFD) analysis is performed to investigate sudden contraction phenomena during operation. The simulation is conducted using Ansys Fluent 14.5 and Maxwell software 2018 R1 (Ansys Company, Canonsburg, PA, USA), Ansys 14.5 Fluent, Ansys Maxwell 2018 R1. The main parameters used in the simulation are as follows: w is the width max value and is 190 mm; $L_{shaft}$ is the shaft length, which is 63 mm, and the resistance value of 2.9 Ohm is used. It is noted that the other design parameters are determined from the simulation to achieve the desired performance of the proposed rotary MR Damper. These parameters are used to analyze the magnetic field distribution. On the other hand, the MR fluid (MRF-132CG, Parker Lord, Cary, NC, USA) is used as the representative nonlinear material, and its nonlinear data for the B–H curve are inputted as engineering data. The coil is made of AWG (American wire gauge) 21 wire with a diameter of 0.75 mm, and the applied current is set at 3 A. The core material is specified as 1008 steel, a ferromagnetic material. The rotational speed is set to 30 m/s. The simulation results of the magnetic field are presented in Figure 6a. Now, based on the magnetic analysis result, the damping forces of the proposed rotary MR damper are calculated at various currents as a function of the rotational speed. Figure 6b shows the simulation results for the magnetic analysis, i.e., the relationship between the field intensity and current and also between the yield stress and current. As the magnetic field strength increases, the yielding stress becomes more pronounced. It can be observed that the magnetic field of approximately 200 kA/m is formed at the pole when a current of 3 A is applied. Consequently, the yielding stress is approximately 40 kPa. This value is corroborated by the red line in Figure 7b.

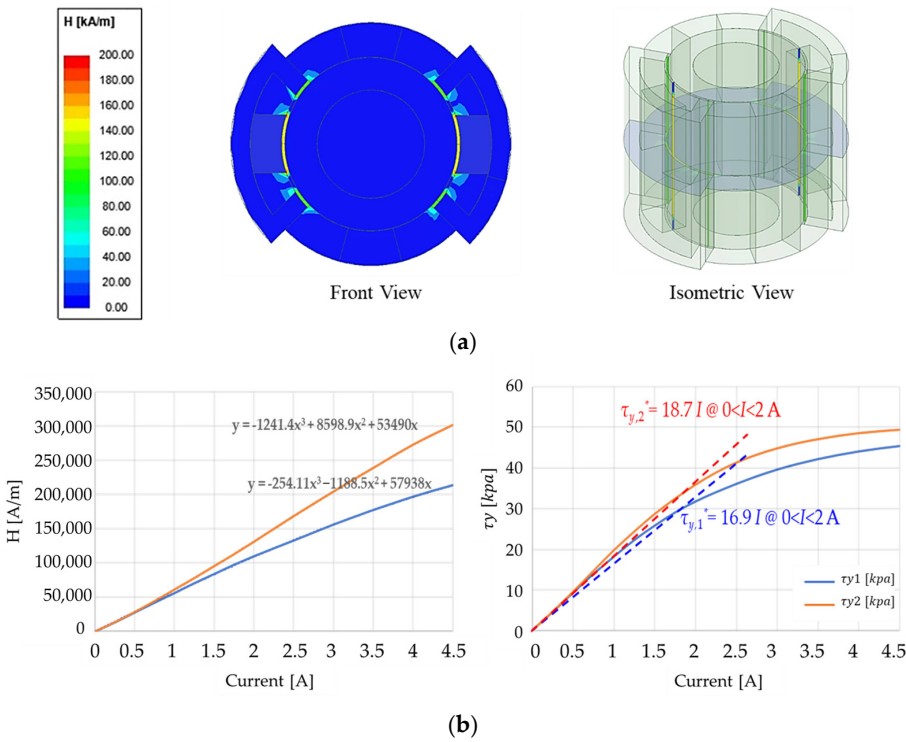

**Figure 6.** Magnetic analysis of the rotary MR damper: (**a**) contour view front, isometric (140 km/A); (**b**) analysis results between the yield stress and current.

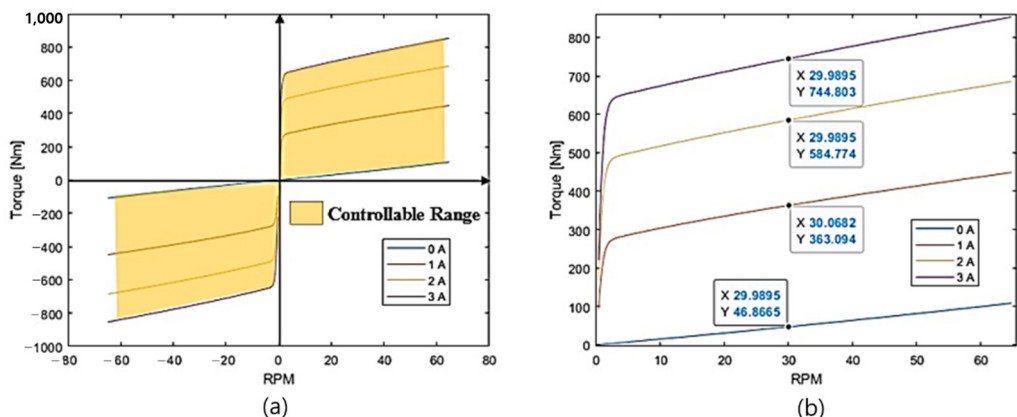

**Figure 7.** Simulation of the field-dependent torque of the rotary MR damper.

For the characterization and performance analysis of the proposed rotary MR damper, torque simulations are conducted at various current inputs and velocity levels. Figure 7 shows the simulation results showing the field-dependent damping torque. Through the simulations, the torque controllable range within the current range of 0 to 3 A and M/S range of −60 to 60 are fixed, as depicted in the Figure 7a. The simulation results exhibited a symmetrical pattern around the origin. Calculations based on positive rpm values indicate torque values of 46.87 Nm at 0 A, 363.10 Nm at 1 A, 584.77 Nm at 2 A, and 744.80 Nm at 3 A, respectively. These values are proportional to the magnetic field. More specific results are summarized as follows: the fluid viscous rotational force (proportional to the rotational speed): 46.87 Nm; shear rotational force (proportional to the magnetic field): 697.9 Nm (3 A input); total rotational force (viscous + shear rotational force): 744.8 Nm (3 A input); outer diameter: 150 mm; width (excluding shaft): 190 mm; rotational angle capabilities: ±30° (capable of rotating a total of 60 degrees); and MR fluid injection volume: 250 mL.

As mentioned in the Introduction, CFD analysis is also carried out in this work to analyze more accurately the pressure drop of the proposed rotary MR damper. It is noted that in a system or device, a pressure imbalance may occur due to the internal rotation or flow of the device. This phenomenon occurs when fluid flows rapidly or solid components move, leading to a rise in pressure. Such occurrences can impact the stability of the system and the durability of its components. As the designed damper experiences real-time changes in internal pressure, it is necessary to examine pressure drop profiles and velocity profiles. Furthermore, comparing pressure drops obtained from mathematical models and CFD simulations helps in the determination. The analysis conditions and simulation results are depicted in Figure 8. The analysis conditions are imposed as follows: The solver type utilized in the simulation is based on the steady pressure. The viscous model adopted is the realizable k-epsilon model, incorporating constraints for C2-epsilon, turbulent kinetic energy (TKE) Prandtl number, and turbulent dissipation rate (TDR) Prandtl number, set at 1.9, 1, and 1.2, respectively. The basic mesh number for both the orifice and the bypass is established at 15, with both regions designated as laminar zones. The solution methods employed include simple pressure–velocity coupling, least square cell-based gradient, standard pressure, second-order upwind momentum, first-order upwind turbulent kinetic energy, and first-order upwind dissipation rate. The residual of absolute criteria is set at $10^{-3}$. Using Equation (7), the maximum Reynolds number and entry length are calculated at 1113 and 87.8 mm when the sink speed is 3.05 m/s and the input current is not applied. At the same time, it is confirmed that the entry length is about 85 mm as a result of the CFD analysis, as shown in Figure 5, performed under the same conditions. Based on Equation (11), the pressure drop calculated by CFD includes both major and minor losses and so includes the total loss coefficients at the orifice and bypass.

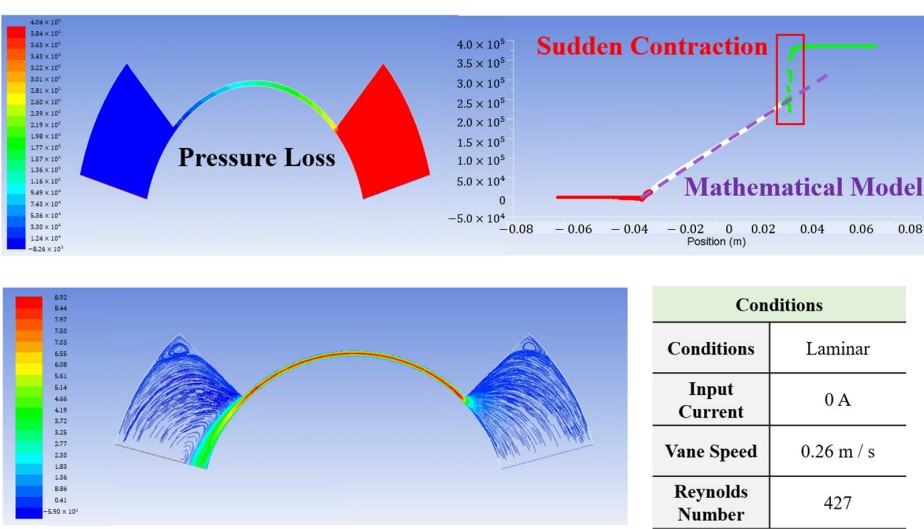

**Figure 8.** Pressure analysis of the rotary MR damper using CFD.

The analysis is conducted under laminar flow conditions, with an input current of 0 A, a flow velocity of 0.26 m/s, and an angular velocity of 60 rpm. The mathematical model yields a total pressure drop of 2.65 bar. The analysis indicates that the pressure drop is relatively low. In contrast, the CFD analysis affects the total pressure drop of 4.04 bar. A comparison of the two values revealed a difference of approximately 1.4 bar. It was concluded that the difference, which is evaluated by 2%, could be considered negligible in the design of the first prototype between two different analysis approaches. In the comparison between the two results, there is a little discrepancy in the rotational damping between the mathematical model and CFD analysis results, especially when the current is not applied (off state). However, the error in rotational damping is deemed negligible compared to the control range between current application (on state) and non-application (off state). Other factors contributing to this deviation may include frictional forces and temperature effects, which are not considered in this work.

## 4. Experimental Results and Discussions

A prototype of the proposed rotary MR damper was manufactured as shown in Figure 9. This prototype is a semi-active damper and applicable to the vehicle suspension systems which have a low floor level of small-sized future cars. In Figure 10, the final assembly covered with an external housing is shown. Assembled within are the shafts that can be connected and attached to the vehicle's axle or suspension. Figure 9b displays the internal core filled with MR fluid. Figure 10 presents an experimental setup for measuring the damping force of the rotary MR damper under the sinusoidal excitations. The damping force is measured by the load cell, while the displacement of the excitation and stroke is measured by the transducer wire sensor. In addition, the servo hydraulic system exciter's damper, current amplifier, and data acquisition system are also used. The experiment was conducted in a consistent environment with a displacement of ±30 mm (rotation ≅ 30°), an excitation frequency of 1 Hz, and a maximum of 15 rpm (0.2 m/s). As seen from the experimental apparatus, the hydraulic exciter originally produced the dynamic motions in a vertical direction only. However, in this test, the vertical motion was converted to a rotational motion using the jigs and fixtures. In order to investigate the effect of the exciting directions from the linear to rotation to the discrepancy, each test was carried out more than five times, and the average values were used for the presentation of the results. It is noted here that the legend of the MR damper stands for the rotary MR damper to be tested with a rotational motion.

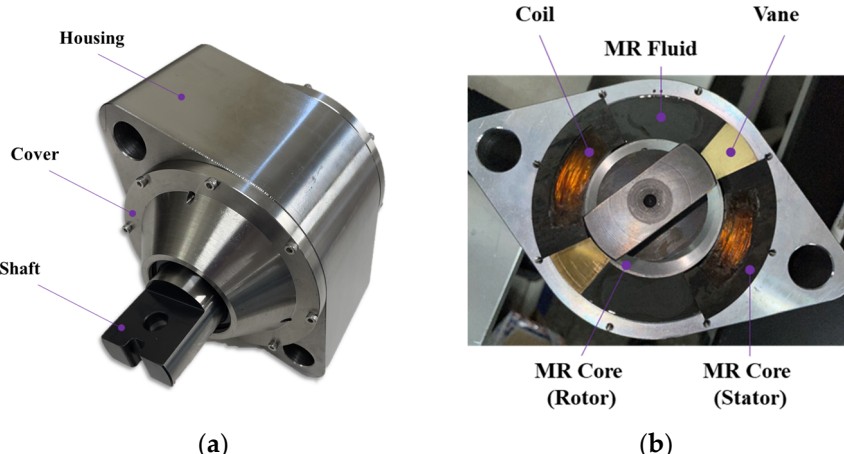

**Figure 9.** The prototype of the proposed rotary MR damper: (**a**) assembled, (**b**) magnetic circuit core.

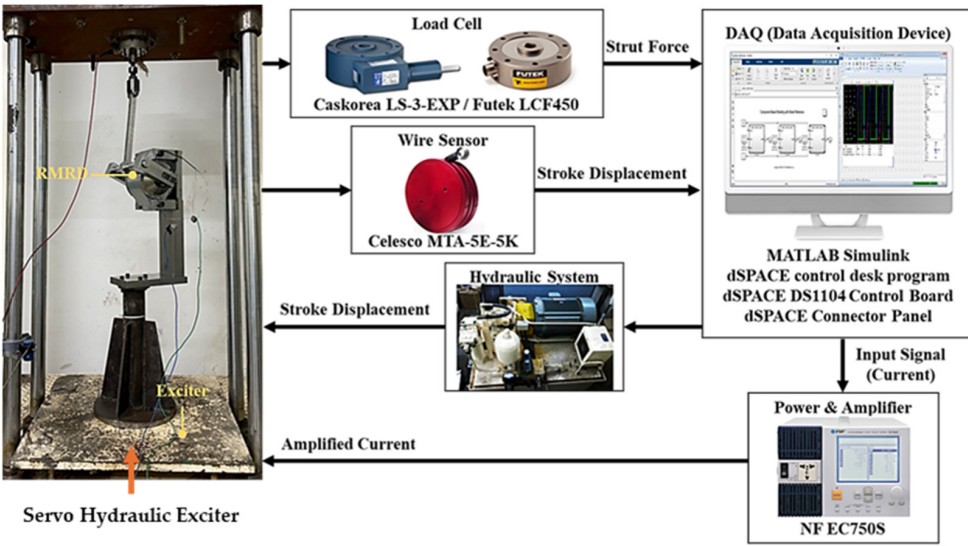

**Figure 10.** Experimental apparatus for the damping force measurement.

### 4.1. Damping Force

Figure 11 shows the field-dependent force achieved from the first test. It was tested in the same way as the linear MR damper by exciting continuously, changing the frequency and magnitude. It is clearly seen that there were severe noises which may have affected the damping force. Moreover, it was found that there was problem in the motion converter. As a result, a large error occurred when compared to the simulation value, so the experimental analysis was conducted immediately. It was determined that the cause of the error occurred in the part where the core inside the damper and the rotor to be contacted produce rolling friction that occurs as the rotor rotates. It was also discovered that when the magnetic field is formed in this area and holds the rotor, then the fluid detours to the contact between the rotor and the core. Due to this phenomenon, the fluid did not accumulate in the designed orifice, but it flowed to an unexpected place, resulting in a lower damping force than expected. It should be noted here that a rotary type of test equipment integrating with the torque sensor and encoder needs to be used to achieve more accurate damping force, angular velocity. However, due to the lack of such test equipment, in this work the low pass filter and lubrication oils were used during the test to compensate for the noise and friction effects. The tolerance that occurred in the motion converter was also reduced before testing again. Figure 12 shows the results after modifying the experimental apparatus. It is noted that in this test, the discrete result at a certain velocity (or rpm) was acquired to make the result point more clearly. The points in the measured results are the tested m/s. Figure 12a presents the simulated force–velocity (F-V) graph, showing hysteresis curves formed in both extension and compression directions for each current. It was found that the force increases or decreases proportionally to the velocity by approximately 1 kN per turn within the velocity range of −0.2 m/s to +0.2 m/s. Thus, it can be inferred that damping force proportional to velocity occurs for input currents beyond a certain level within the block-up region. The measure of the damping force is shown in Figure 12b, achieved from the same operating conditions except the data acquisition method, where the discrete damping force was collected at a certain one m/s condition. It is identified from two results that the maximum damping force achieved from the experiment was little less than the simulated result at the same piston velocity. It should be noted here that the maximum force shown in Figure 12b at 0.2 m/s is almost equivalent to the desired damping torque of 600 Nm.

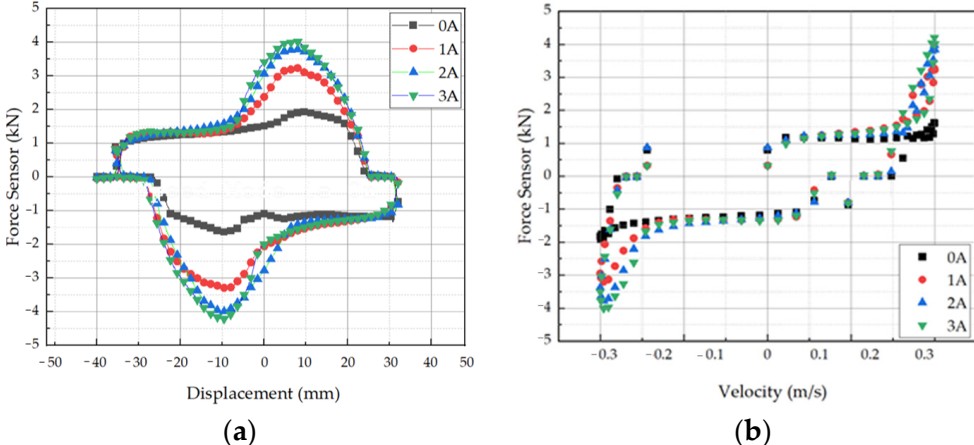

**Figure 11.** Damping force experiment results data, with simulation data reapplied with 0.2 m/s 0~3 A: (**a**) F–D curve for 0.2 m/s, (**b**) F–V curve for 0.2 m/s.

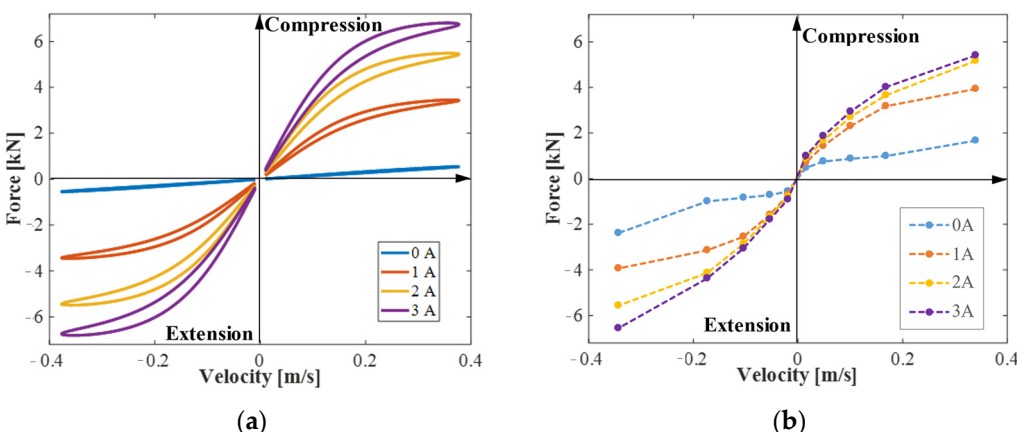

**Figure 12.** Damping characteristics in F–V curve: (**a**) simulation, (**b**) measurement.

### 4.2. Response Time Measurement

One of the salient properties of MR fluid is the fast reversible response time, which can be adaptable to various dynamic systems subjected to relatively high frequency disturbances. As a vehicle suspension system, there are two significant vibration modes: first mode (car body mode) and second mode (wheel mode). In order to cover both modes, the response time of the suspension system should be faster than 50 ms at least. Therefore, the response time of the MR damper itself needs to be faster than 30 ms to avoid the adverse effect from the time delay. In general, in order to achieve the fast response time of the MR damper, several methods have been carried out, for example, the use of the register in the magnetic circuit, the reduction in the eddy current, the surface groove of the magnetic pole, and others. In this work, a simple PI control circuit is used to achieve a fast response time of the rotary MR damper. Figure 13 shows the configuration of the current control PI controller integrated with the MR damper, in which the PI controller can help improve the response time of the overall control looping time. Table 1 shows the measurement results of the response time of the proposed MR damper with the PI circuit shown in Figure 13. Since the inductance increases as the current increases, and hence the impedance is expected to decrease. This reaction speed is expected to improve when a high current is applied. When constant voltage is applied instead of the PI controlled voltage, there is very little difference in rising time within 10 ms. It is also observed from the results that the rise time becomes longer when the 60 Ohm register is added to the PI circuit. The finding of the solution to resolve this problem takes time, and hence it is not treated in this work. However, it is surely found that the rise time in most of the cases is less than 50 ms, which is the target value. Therefore, the prototype rotary MR damper, which is applicable to the low-body floor vehicle suspension system or PBV suspension system, can be a novel candidate for the suspension system of special vehicles to be appeared in UMC vehicles.

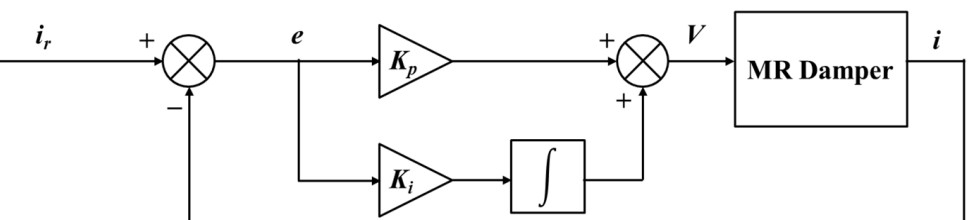

**Figure 13.** PI control block diagram to achieve fast response time of the rotary MR damper.

**Table 1.** Measured response time of the rotary MR damper evaluated at 90% of the rise time.

| Type | $K_p$ | $K_i$ | Rise Time (up, Cycles) | Rise Time (ms) | Remark |
|---|---|---|---|---|---|
| 0 A → 3 A | 2 | 10 | 12 | 22 ms | |
| | 2.5 | 10 | 11 | 20 ms | |
| | 3.5 | 10 | 11 | 20 ms | |
| | 1 | 10 | 16 | 30 ms | |
| | 0.5 | 10 | 18 | 34 ms | |
| | 0.5 | 5 | 20 | 38 ms | |
| | 2.5 | 5 | 14 | 26 ms | |
| | 2.5 | 1 | 13 | 24 ms | |
| | 2.5 | 15 | 13 | 24 ms | |
| | 4.0 | 10 | 16 | 30 ms | |
| | 2.5 | 0 | 14 | 26 ms | |
| | 5 | 0 | 14 | 26 ms | |
| 0 A → 1 A | 2.5 | 10 | 79 | 156 ms | Abnormal behavior expected |
| | 5 | 0 | 82 | 162 ms | |

## 5. Conclusions

In this study, a new type of rotary MR damper applicable to the low-body floor vehicle suspension system was designed, and its effectiveness was demonstrated through the testing of the prototype. In the design process, the targets of the design parameters and maximum damping force were first specified, unlike other research works. After formulating the governing equations of morion in a non-dimensional domain, the pressure drop and flow motion were analyzed, followed by magnetic field analysis through both the finite element method and CFD approach. Therefore, principal design parameters to achieve the prescribed targets have been determined, and the field-dependent damping force is achieved in the force versus velocity (rotational speed). After validating from the simulation results that the target specifications could be obtained using the determined design parameters, a prototype of the rotary MR damper was manufactured and tested. It was confirmed that the target maximum damping force of 600 Nm could be achieved from the prototype and the target rise response time of 50 ms also could be easily obtained by integrating a simple PI electric circuit. It is finally noted as future works that some benefits to be expected from the proposed rotary MR damper should be deeply explored by focusing more on compact size, a small amount of MR fluid, fewer sealing problems, low energy consumption, fewer particle settling problems due to the rotation motion, a higher damping force through optimization, and longer durability for practical feasibility. In addition, the temperature and eddy current effects on the suspension performance need to be undertaken through a quarter suspension model in which the proposed MR rotary MR damper is installed.

**Author Contributions:** Conceptualization, B.-H.K. and S.-B.C.; methodology, B.-H.K.; software, B.-H.K.; mathematical modeling, B.-H.K., validation, B.-H.K.; formal analysis, B.-H.K.; investigation, B.-H.K.; data curation, B.-H.K.; writing—original draft preparation, Y.-J.P.; writing—review and editing, S.-B.C.; visualization, B.-H.K. and Y.-J.P.; supervision, S.-B.C.; project administration, S.-B.C. All authors have read and agreed to the published version of the manuscript.

**Funding:** This work was partially supported by Hyundai Motors via the research program of "The concept development on MR damper for active suspension of e-corner module".

**Data Availability Statement:** Data are contained within the article.

**Conflicts of Interest:** The authors declare no conflicts of interest.

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
