# Peer review of "A New Rotary Magnetorheological Damper for a Semi-Active Suspension System of Low-Floor Vehicles"

_actuators, doi:10.3390/act13040155_

Round 1

Reviewer 1 Report

Comments and Suggestions for Authors

Actuators ID paper: 2945259

Article Title: A New Rotary Magnetorheological Damper for a Semi-Active  Suspension System of low-Floor Vehicles

The aim of the Journal Actuators is to present and discuss on the scientific research, technology development, engineering, and the science and technology states of actuators and their control systems in the last period of time. The proposed techniques and applications should be innovative and significant enough for the community interested of this area of knowledge. The considered paper deals with a new type of Magnetorheological Damper of a Semi-Active Suspension System for low-Floor Vehicles. In order to address the limitations of conventional linear Magneto-Rheological (MR) dampers in controlling vibrations in vehicles with lowered chassis, in this study, Authores have proposes the use of rotary MR dampers. The considered damper, in the paper, is designed through mathematical modeling and magnetic field analyses, and a prototype is fabricated for experimental validation. The experimental results - presented in the paper - demonstrate the effectiveness of the proposed MR damper. 

I won't hide that the proposed structure of the MRF damper and its operating principle described in the article interested me greatly. However, delving deeper into the content, I noticed a significant lack of essential information. But let's start from the beginning. The paper has a very strong and interesting introduction. The description of the system (Chapter 2) is also engaging. However, a weak point here is Figure 3. Out of the two depicted systems, only one (3b) is given in more details. Moving on, Chapter 3 concerning the electromagnetic calculations of the considered damper is quite limited. While the authors do reveal information about the current values powering the system and the turns number of windings used, there is no mention of the lumped parameters such as resistance or inductance - these parameters heavily influence the response time of the system. My doubts also arise from the use of commercial software for analyzing field phenomena, including magnetic field distribution or fluid flow field. From my own experience, I know how many additional steps are required to align the calculations obtained in commercial programs with reality. Did the authors consider the possibility of applying their own programs directly correlated to the specific field? Furthermore, I did not notice any information on whether, for example, the field calculations were performed separately as coupled. In both cases, I also miss basic information such as initial conditions in the conducted simulations. Additionally, the presented distribution of the magnetic field intensity vector is difficult to read. It's hard here to find the value of 200 kA/m mentioned by the authors. In my opinion, the average value of the H vector fluctuates at most in the range of 120-140 kA/m. It would be good practice to present an algorithm illustrating the derivation of, among other things, the curve from Figure 6b, as well as other calculation results. I would also like to note that the quality of Figure 8 could be much better. One question that still puzzles me in the analysis of the considered system is whether the authors considered the influence and effect of eddy currents in the studied system - undoubtedly, the majority of damper components are 'solid' type. There could be many more questions posed in the space of EM field modeling and CFD. One of them is, among others, why the influence of temperature was omitted in the design calculations? I believe that the chapter devoted to modeling should be thoroughly redesigned to highlight the essential issues related to analyse phenomena.

Nonetheless, I must say that I really like the concept of the system as I mentioned at the beginning.

Reviewer 2 Report

Comments and Suggestions for Authors

The main contribution of this work is to design and manufacture the prototype of the rotary MR damper and to demonstrate its effectiveness to the possibility of practical application for future vehicles with very low floor height.

1, Not all equations from (1)… are numbered. Each equation should be numbered.

2. There is no specification for the parameters h, w, c, etc.

There is an undefined function yield stress τy (H) in equation (2).

3. Below each equation should be a list of parameters, followed by "where: Q is the".

4. "Dynamic viscosity" or "kinematic viscosity" should be clearly specified.

5. Without a list of all parameters used in the model equations, it is impossible to verify their correctness. It is recommended to include a list of model parameters and its values in the appendix.

6. In Equation (6) the Nb parameter is unidentified.

7. The diagram of the servohydraulic exciters for the damper is not shown in Figure 10.

8. There is no comparison of simulation and measurement damping characteristics F-V in one figure; then you can see the differences better.

9. When MR is exciting with an external force of 1 Hz, the MR frequency responses should also be analyzed.

10. No evaluation of the effectiveness of the PI controller. Only the measured response time of the rotary MR damper was compared. The gain of the PI controller should guarantee system stability, minimize the response time of the system, and ensure robustness against errors.

Round 2

Reviewer 1 Report

Comments and Suggestions for Authors

The authors responded to my comments and made appropriate changes to the article.

Reviewer 2 Report

Comments and Suggestions for Authors

The authors responded to all comments, which is sufficient at this stage of work